# Cerebellar High-Grade Glioma: A Translationally Oriented Review of the Literature

**DOI:** 10.3390/cancers15010174

**Published:** 2022-12-28

**Authors:** Ashley L. B. Raghu, Jason A. Chen, Pablo A. Valdes, Walid Ibn Essayed, Elizabeth Claus, Omar Arnaout, Timothy R. Smith, E. Antonio Chiocca, Pier Paolo Peruzzi, Joshua D. Bernstock

**Affiliations:** 1Department of Neurosurgery, Brigham and Women’s Hospital, Harvard Medical School, Boston, MA 02115, USA; 2Oxford Functional Neurosurgery Group, Nuffield Departments of Surgical Sciences, University of Oxford, Oxford OX3 9DU, UK; 3Department of Neurosurgery, Boston Children’s Hospital, Harvard Medical School, Boston, MA 02115, USA; 4Department of Neurosurgery, University of Texas Medical Branch, Galveston, TX 77555, USA; 5David H. Koch Institute for Integrative Cancer Research, Massachusetts Institute of Technology, Cambridge, MA 02139, USA

**Keywords:** cerebellum, glioblastoma (GBM), high-grade glioma (HGG), immunotherapy, experimental therapeutics

## Abstract

**Simple Summary:**

High-grade glial cancers typically arise in the cerebral hemispheres and only rarely elsewhere in the brain. Historically, such tumors arising in the cerebellum have been handled clinically as per their cerebral counterparts. However, recent epidemiological research and molecular analyses have demonstrated that these tumors are different in ways that are likely to be relevant with regard to therapeutic intervention(s). Accordingly, this review charts the landscape of this evidence and highlights emerging translational opportunities for treatments of high-grade cerebellar gliomas.

**Abstract:**

World Health Organization (WHO) grade 4 gliomas of the cerebellum are rare entities whose understanding trails that of their supratentorial counterparts. Like supratentorial high-grade gliomas (sHGG), cerebellar high-grade gliomas (cHGG) preferentially affect males and prognosis is bleak; however, they are more common in a younger population. While current therapy for cerebellar and supratentorial HGG is the same, recent molecular analyses have identified features and subclasses of cerebellar tumors that may merit individualized targeting. One recent series of cHGG included the subclasses of (1) high-grade astrocytoma with piloid features (HGAP, ~31% of tumors); (2) H3K27M diffuse midline glioma (~8%); and (3) isocitrate dehydrogenase (IDH) wildtype glioblastoma (~43%). The latter had an unusually low-frequency of epidermal growth factor receptor (EGFR) and high-frequency of platelet-derived growth factor receptor alpha (PDGFRA) amplification, reflecting a different composition of methylation classes compared to supratentorial IDH-wildtype tumors. These new classifications have begun to reveal insights into the pathogenesis of HGG in the cerebellum and lead toward individualized treatment targeted toward the appropriate subclass of cHGG. Emerging therapeutic strategies include targeting the mitogen-activated protein kinases (MAPK) pathway and PDGFRA, oncolytic virotherapy, and immunotherapy. HGGs of the cerebellum exhibit biological differences compared to sHGG, and improved understanding of their molecular subclasses has the potential to advance treatment.

## 1. Introduction

Cerebellar high-grade gliomas (cHGG) (WHO grade 4) account for ~1% of central nervous system HGG [1]. This is disproportionately less frequent relative to the number of glial cells in the cerebellar parenchyma as compared to their supratentorial counterparts [2], and suggests a relative biological resistance for cerebellar cells to undergo malignant transformation. Indeed, over past decades the primary question regarding cHGG has been: in what way are cHGGs biologically different as compared to supratentorial high-grade gliomas (sHGG)? Unfortunately, treatments have not differed between cHGG and sHGG and neither have outcomes been obviously different. However, entering a time of genomically targeted treatments, in which the uniformity of tumors is conceptually discarded and each malignancy is considered as its own disease, the unique molecular characteristics of cHGG have come into perspective. Recognizing this, we present a review on the epidemiology, molecular characteristics, and treatment of cHGG as new therapeutic avenues diverge from sHGG.

### 1.1. Clinical Epidemiology

Up to the mid-20th century, the very existence of cHGG was questioned, due to its low incidence [3], and likely due to suboptimal visualization of posterior fossa structures by CT. Subsequent estimates of prevalence in this region are that approximately 1% of all HGG are principally cerebellar in location (Figure 1) [1]. A small number of modern papers provide relevant epidemiological information comparing sHGG to cHGG (Table 1). Major differences in male-female ratio and overall survival are not evident; however, the average age is lower for cerebellar patients. For example, analysis of the US Surveillance, Epidemiology, and End Results (SEER) registry—offering the largest adult cohort of these tumors—finds 36% of patients are over 65 and 24% below 40 years-of-age, compared to 46% and 7%, respectively, in supratentorial locations [4]. Although multifocality has been observed at a high rates (21–33%) in cHGG [5,6], other reports find a similar rate to sHGG [7], where an incidence of 5% is typical. Greater overall survival at 2 years has been reported for cHGG (e.g., 22 vs. 8%) [4]. However, this is controversial as other authors report worse prognosis (e.g., 12 vs. 32% at 2 years) [5]. This finding may be a composite of an increased propensity for brainstem invasion but less aggressive tumor progression, which may partially offset each other [6,8].

### 1.2. Molecular Characterization and Features

The rarity of HGG in the cerebellum has stimulated authors to reflect further on biological differences. Several theories have been advanced to explain this relative resistance to gliomagenesis in the cerebellum, such as the depletion of substance P in the adult cerebellum [11]. Some have even argued that most of these tumors are not genuinely cerebellar in origin, but instead, represent metastases from an occult supratentorial or brainstem site [12]. However, there is no robust evidence for this hypothesis. Recent molecular evidence, namely the absence of the FOX1 telencephalic marker and other signature gene expression, clearly shows that cHGGs are genuinely cerebellar and not cerebral in origin [10]. The relative resistance to typical pathways of gliomagenesis and rarity of cHGG suggests that alternative pathways may have greater importance among cHGG compared to sHGG, a hypothesis which is now being borne out in the data.

Historically, it was believed that primary HGGs or glioblastomas (GBM) arose de novo as grade 4 lesions, whereas secondary HGGs (currently, removed from the category of GBM) could be traced clinically to a prior lower grade tumor. Compared to primary sHGG, secondary sHGGs are characterized by a high frequency of p53 mutations, absence of epidermal growth factor receptor (EGFR) amplification, a bias toward younger patients, [13,14] and in particular, isocitrate dehydrogenase (IDH) mutations [15]. Clinically, secondary HGGs are identifiable in approximately 10% of supratentorial and cerebellar cases [6,7,15]. However, molecular genetic evidence has revealed that cHGGs have a mix of features of both primary and secondary HGG (defined by the supratentorial molecular framework); IDH mutation is uncommon (primary feature—although rare mutations are typically not tested) as is EGFR amplification (secondary feature), and a moderate frequency of p53 mutations (Figure 2) [16]. Certainly, the near absence of EGFR mutations is in contrast to the supratentorial population of tumors [17,18]. The regular observation of a mixed primary–secondary pattern in cHGGs led authors to postulate that these cerebellar tumors tend to develop through a different collection of processes or pathways to their supratentorial comparators [19,20,21]. This now seems well justified, and indeed, compared to sHGG, there are many genetic differences between the two cohorts. For example, there is a higher incidence of neurofibromatosis 1 (NF1) mutations [5,10], unusual RAS mutations [10,16], and a larger population of H3K27M mutated tumors within this cohort [5], otherwise known as grade 4 diffuse midline glioma (DMG) [22,23,24].

Detailed molecular analysis on histological cHGG has been performed by three groups: in Seoul [10], Heidelberg [25], and Tokyo [26]. This research, particularly the work of Cho et al. and Reinhardt et al. which both provide a supratentorial comparator [10,25], is definitive in re-casting cHGG as a meaningfully separate tumor population to sHGG. As a population, they are most similar to proneural-sHGG, and upregulation of genes such as SOX 10, CSPG4, and OLIG2 strongly suggest a dominant oligodendrocyte lineage [10,26]. Interestingly, topological transcriptome and DNA methylome cluster analysis locates cHGG within the sHGG cluster but polarized toward the pilocytic astrocytoma cluster—a predominantly cerebellar tumor [10]. These malignant cerebellar tumors express a set of signature genes and are populated by a set of HGG subclasses that may each merit a different therapeutic approach (Box 1). Most notably, these include high-grade astrocytoma with piloid features (HGAP) [22,27,28], H3K27-altered DMG, and particular methylation subclasses (midline and RTK I) of IDH-wildtype GBMs (Figure 3). Ultimately, what was initially observed in cHGG as a mixed primary–secondary genetic pattern was in fact the manifestation of the predominance of these HGG subclasses. Comparison of pediatric and adult cHGG is tenuous due to sparse data but DMG H3K27 and GBM IDH-wt subclass midline may be more prevalent and HGAP may be less prevalent than in adults (Figure 3) [25].

Box 1Cerebellar high-grade glioma population features.
IDH mutation is uncommonEGFR amplification is rareSubstantive HGAP subpopulation (~1 in 3)Substantive DMG H3K27 subpopulation (~1 in 10)More frequent RAS mutations, ATRX alteration, PDGFRA amplification, CDK2A/B loss, and CDK4 amplification than sHGGLess frequent TERT promotor mutations than sHGGMethylation classes predominantly: o(1) high-grade astrocytoma with piloid features (HGAP),o(2) GBM IDH-wildtype subclass midline (GBM-MID), o(3) GBM IDH-wildtype subclass RTK I,o(4) diffuse midline glioma H3K27-altered (DMG H3K27).Scant expression of telencephalic marker (FOX1)Widespread expression of cerebellar marker (PAX3)Widespread expression of oligodendrocyte progenitor marker (CSPG4)


## 2. Therapeutic Approaches

### 2.1. Classic Therapy

Therapy for cHGG has not meaningfully deviated from that for sHGG: maximal resection followed by radiotherapy and, since 2005, concurrent chemotherapy with temozolomide (TMZ). Adjuvant TMZ or carmustine chemotherapy regimens are also often employed. While there is no class 1 evidence to support this approach specifically for cHGG, some retrospective cohort analyses have corroborated the efficacy of radiotherapy and surgical resection. Namely, Weber et al. show an association of additional treatment after surgery with longer survival [6]; the analysis of SEER by Babu et al. shows an association of resection (8 vs. 4 months) and radiotherapy (11 vs. 3 months) with longer survival [9]; and Yang et al. show an association of radiotherapy (15 vs. 6 months) and degree of resection (15 vs. 6 months) with longer survival [8]. Despite TMZ being considered part of the gold-standard of treatment, direct cohort-based evidence for efficacy of current chemotherapy is limited to one series demonstrating a modest benefit that did not reach statistical significance [29]. However, it is perhaps relevant that since its institution, median overall survival has tended to be longer (Figure 1C, note that SEER includes patients from 1973 with only 35% diagnosed since 2005, and Picart et al. [5] reports a 35% rate of TMZ use). There remains a lack of consensus on whether radiotherapy should be delivered locally, to the whole brain, or with a craniospinal distribution. For example, some authors cite craniospinal treatment as important for minimizing metastasis [30]. It has been postulated that radiotherapy is particularly relevant to cHGG given the high frequency of unamplified EGFR [20]. The grounds for this assertion are the association of radio-resistance and EGFR^+^ in sHGG [31,32], and the anecdotal finding of longer survival of radiotherapy-treated EGFR^-^ cHGG [20].

### 2.2. Emerging Molecular and Cellular Therapies

Translational approaches to cHGG cover a range of therapeutic classes and are specific to the properties of the tumor subclass (Box 2).

Box 2Translational approaches for cerebellar high-grade glioma.
cHGG is likely comprised of different proportions of distinct molecular subclasses compared to sHGG (GBM IDH-wt, HGAP, and DMG).Distinct approaches are engaged for major subclasses: oHGAP: MAPK kinase inhibitors, PI3K/mTOR inhibitors, cyclin-dependant kinase inhibitors, ATR inhibitorsoDMG: GD2-CAR T cell therapyoGBM IDH-wt: PDGFRA inhibitors, cyclin-dependant kinase inhibitors, combination immunotherapiesUnfavourable GBM IDH-wt immunological environment hampers many immunotherapiesOncolytic viruses are likely cerebellum-safe and promote favourable immunological environment, opening a gate for additional therapies.Oncolytic virus-based multi-modal immunotherapy is an attractive strategy for cerebellar GBM IDH-wt.Greater knowledge of cerebellar glioblastoma immunology is needed, particularly HGAP and GBM IDH-wt.


#### 2.2.1. High-Grade Astrocytoma with Piloid Features

HGAP is a recently defined, IDH-wt glioma that predominantly originates in the cerebellum and is a large and important subclass of cHGG (Figure 3) that expresses a number of features that advocate for a targeted approach [28]. The low-grade pilocytic astrocytoma (PA) is a common tumor preferentially effecting the cerebellum in children. It is essentially a single pathway disease that involves a BRAF-KIAA1549 fusion, particularly when the tumor is located in the cerebellum [33,34]. HGAP was initially classified from a unique methylation signature from cases of PA with anaplastic histological features that exhibited more aggressive behavior [28]. While PA rarely transforms to HGG, particularly when the BRAF fusion is present [35], a considerable number of HGAP (~20%) have been identified with this fusion, which suggests an origin secondary to prior PA or at least some biological similarity [25,36]. This fusion abnormally drives the mitogen-activated protein kinase (MAPK) pathway. Indeed, up to 75% of HGAP possess either this fusion or other MAPK pathway alterations, such as NF1 mutation/deletion, fibroblast growth factor receptor (FGFR) mutation/fusion, KRAS mutation, or BRAF-V600E mutation [28]. Drug testing of subclass-undifferentiated cHGG demonstrates a higher sensitivity of cHGG to MAPK kinase inhibitors (MEKi) than sHGG [10]. This can presumably be attributed to the considerable representation of HGAP among cHGG. In PA BRAF-KIAA1549 fusion models, a credible role for RAF inhibitors, such as PLX8394 [37], or MEKi, such a trametinib, has been demonstrated [38]. Escape to these agents develops via the PI3K/mTOR pathways, and as such can be frustrated with mTOR inhibitor dual-therapy, such as everolimus [38,39]. As NF1 is a negative regulator of RAS, RAS inhibitors such as tipifarnib or downstream MEKi may be of therapeutic value. Many FGFR tyrosine kinase inhibitors are available, and indeed PD173074 has shown in vitro efficacy in retarding HGG growth [40]. Alpha thalassemia/mental retardation syndrome X-linked (*ATRX*) is a gene involved in telomere maintenance, and loss/mutation is observed in ~45% of HGAP [28]. Tumors harboring this alteration have been noted to have greater sensitivity to DNA damaging agents, such as TMZ [41]. Failure of normal telomere maintenance, via compromise of ATRX, is associated with an alternative lengthening of telomeres (ALT) pathway via a recombination-based process, thereby overcoming replicative mortality. However, inhibition of protein kinase ATR—a regulator of the ALT process—by VE-821 disrupts this pathway and triggers apoptosis [42]. Lastly, the tumor suppressors cyclin-dependent kinase inhibitor 2A and B (*CDKN2A/B*) are found to be deleted/mutated in ~80% of HGAP. Palbociclib is a cyclin-dependent kinase inhibitor, and in such altered gliomas it may have therapeutic potential, particularly in the ‘proneural subclass’ [43]. This is the gene set which is highly enriched in cHGG [10]. A phase-II clinical study failed to demonstrate benefit of palbociclib in recurrent HGG [44], but in light of pre-clinical evidence that concurrent radiotherapy is required for efficacy [45,46], further clinical study is warranted, including in HGAP-cHGG.

#### 2.2.2. Diffuse Midline Glioma, H3K27M

The defining feature of DMG-H3K27M is abnormal histone modifications leading to epigenetic derangements [22,23]. Preclinical work has investigated histone deacetylase inhibition and histone demethylase inhibition [47], and immunologic targeting of the mutated histone 3 [48], which are entering early-stage clinical trials. The disialoganglioside GD2 is highly expressed by DMG-H3K27M cells, which has been utilized as the basis of chimeric antigen receptor (CAR) T-cell therapy in pontine and spinal cord locations [49,50]. This comprehensive phase I report underlines the promise of this therapy, which could extend to cerebellar DMG. The STAT3 transcription factor has been found to be highly upregulated in DMG and high relative expression is associated with shorter patient survival [51]. Inhibition of this pathway with the kinase inhibitor WP1066 results in stasis of tumor growth, confirming its potential as an avenue of treatment.

#### 2.2.3. Glioblastoma, IDH-wt

Emerging treatments targeted to cerebellar GBM IDH-wt can in general be considered alongside the majority of primary sHGG and are reviewed elsewhere [52,53]. While these populations are largely comparable, they demonstrate less frequent EGFR amplification and more frequent CDKN2A/B loss and PDGFRA amplification than the supratentorial entities [25], as well as enrichment of PDGFRA-associated genes [26]. Consistent with this, in vitro, molecules targeted to inhibit PDGFR, namely tivozanib and tandutinib, have shown a greater impact on cell viability in cHGG than sHGG [10]. Similarly, inhibitors targeted to EGFR were substantively less effective in cHGG. As such, these examples serve as a caveat to transposing emerging treatments from supratentorial to cerebellar IDH-wt GBM. Treatments targeted to or favored by the proneural subclass or either midline or RTK I methylation classes of IDH-wt GBM are probably most suitable for translation to cHGG.

#### 2.2.4. Immunotherapies

Immunotherapies have already led to dramatic successes for a range of hematological and solid malignancies. This extent of response has not been reproduced in brain malignancies such as HGG. This is attributed in large part to the dearth of tumor-associated T cells, typically described as immunologically ‘cold’ [54]. Furthermore, lymphotoxic TMZ and potent corticosteroids used to control edema likely contribute to blunted immune activation. These tumors are also characterized by great intratumoral heterogeneity [55,56,57], and ability to evolve antigen escape [58]. As such, a plausible strategy involves an attempt to (1) convert HGG to a ‘hot’ tumor with an expanded and active lymphocytic compartment, and (2) deliver combinations of immunotherapies, leveraging the immune system to eliminate malignant cells and organized to close down avenues of clonal escape.

Oncolytic immunovirotherapy is a potentially attractive platform to achieve these goals, utilizing neurotropic neurovirulence-attenuated viruses, such as oncolytic herpes simplex 1 (oHSV) G207 [59]. In addition to direct oncolysis, they can markedly alter the tumor microenvironment [60] in a way that is likely to promote susceptibility to other immunotherapies such as CAR T-cells, checkpoint inhibitors, and dendritic cell-based vaccines (DCV). In consideration of cHGG, our group has established pre-clinical safety of cerebellar inoculation of oHSV-G270 [61], demonstrated safety and immunological response in pediatric sHGG [62], and are carrying out a phase I study in recurrent pediatric cerebellar tumors including cHGG. This trial will prove to be the first ever use of an oncolytic virus in the cerebellum [63].

Immune checkpoints serve as a physiological mechanism of self-tolerance but are utilized by malignancies to evade lymphocytic attention. Immune checkpoint inhibitors (ICIs) work to stymie this process by handicapping the agents promoting this mechanism (most notably CTLA-4 and PD-1), thereby unencumbering the immune system to treat the malignant cells as foreign. Despite promising pre-clinical work, phase III trials of PD-1 checkpoint blockade (Checkmate 134 [64], 498, and 548) have not yet yielded success in sHGG, although other trials are ongoing (see Mende et al., 2021 for summary) [65], and neoadjuvant, as opposed to adjuvant, timing may be favorable [66]. The tumor’s immunological microenvironment is recognized as a major factor in response to ICI therapy [66,67], so ultimately cHGG is likely to require immunological priming before ICIs can play a decisive role [68]. Oncolytic viruses offer an apposite pairing, with exciting pre-clinical evidence to support this concept of use [69,70]. Little is known about the differential characteristics of HGAP immunology, therefore at present conclusions drawn from sHGG are not directly commutable. DCVs comprise autologous DCs, matured ex vivo with tumor-specific antigens, which can then be re-introduced intradermally with the aim of educating T cells to recognize tumor epitopes as foreign. Preliminary median overall survival results of a phase III trial are promising but remain blinded at present, therefore requiring equipoise [71]. Similarly to ICI, this therapy may well be a complementary companion to oncolytic immunovirotherapy [68]. Surgical resection and viral treatment of the tumor bed can be followed by processing of the tumor sample as part of generating a DCV, which can then be introduced to an immunologically ‘hot’ tumor environment.

The ‘cold’ tumor microenvironment is comprised of abundant myeloid-derived suppressor cells and regulatory T cells, and low numbers of activated lymphocytes and NK cells [67]. Indeed, 30–50% of HGG cells are macrophages or microglia [72,73]. Tumor and macrophages have a complex relationship (see Yu et al., 2021 for summary [53] and Andersen et al., 2021 for in-depth review [74]), but ultimately generate an anti-inflammatory and pro-tumorigenic condition that is hostile to treatment. Arguably the most investigated example of targeting this compartment is by CSF-1 inhibition, which is particularly amenable to the ‘proneural’ subclass (to which most cerebellar GBM IDH-wt belong) [75]. However, resistance develops quickly via the PI3K pathway [76], and clinical evaluation has failed to show efficacy [77]. A number of other targets, such as STAT3 and IDO1 are the subject of current clinical trials [74].

The crucial role of the tumor immunological microenvironment in treatment resistance of HGG is well-established, but also identifies alternative avenues [74]. This understanding is mostly borne from the study of DMG and supratentorial or supratentorial-type HGG [74,78]. While many mechanisms and features are shared, important differences exist between IDH-mut, IDH-wt, and DMG. Furthermore, genes such as NF1 that regulate the immune microenvironment are differentially altered in cHGG and sHGG populations [5,10,79]. It follows from this, to best rationalize and select prospective immunotherapies for cHGG tumors, cultivating a specific understanding of cerebellar GBM IDH-wt and HGAP microenvironment immunology is a sensible endeavour.

## 3. Conclusions

The cerebellum appears to be a privileged site that is resistant to developing HGG, with cHGG harboring different molecular characteristics compared to sHGG. The treatment of cHGG is beginning to diverge with the recognition that a substantial portion of these tumors may actually represent HGAP or DMG-H3K27M. Further understanding of the mechanisms of gliomagenesis in this region and of targeted treatments toward the molecular drivers of individual tumors and their immunologic milieu will be the next step toward personalized and efficacious treatment options for cHGG.

## Figures and Tables

**Figure 1 cancers-15-00174-f001:**
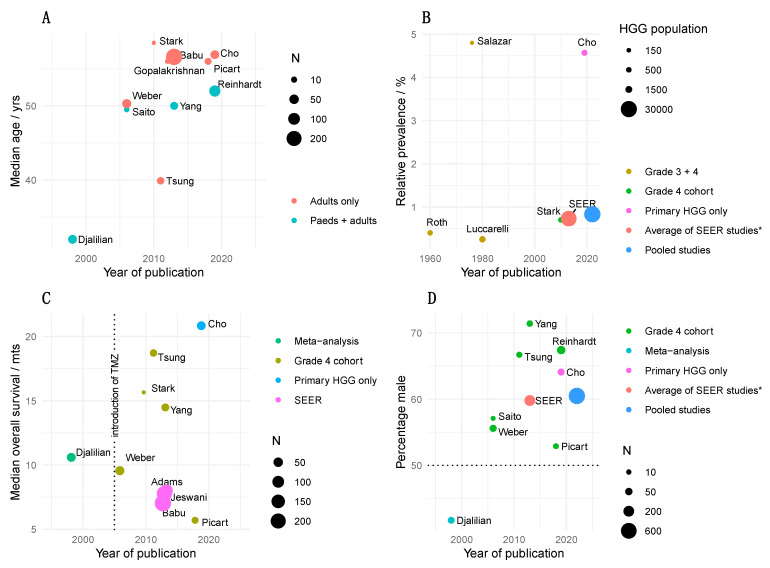
Epidemiology of cerebellar high-grade glioma. Reports are plotted against publication date and identified by the first author. (**A**) median age. (**B**) percentage of high-grade glioma tumors which are cerebellar. (**C**) median overall survival. (**D**) percentage of male patients. Older reports where high-grade glioma explicitly includes grade 3 and 4 tumors are noted. For (**B**,**D**), a pooled average (sky blue) of the reports is given at year 2022. N = number of patients; SEER = Surveillance, Epidemiology, and End Results; HGG = high-grade glioma. * Weighted average of Adams et al. [1], Jeswani et al. [4], and Babu et al. [9].

**Figure 2 cancers-15-00174-f002:**
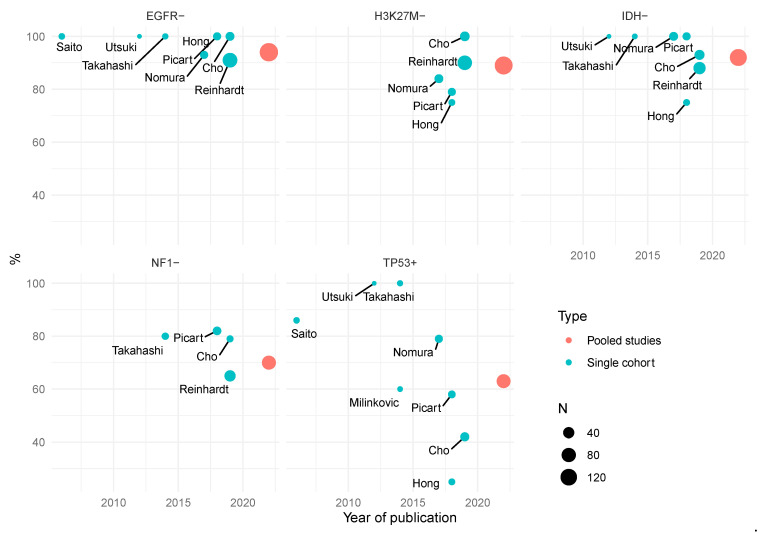
Frequency of gene alteration in cerebellar high-grade glioma. Reports (blue) are identified by first author and plotted by both publication date and the percentage gene prevalence in the respective cohorts. For each gene, a pooled average (red) of the reports is given at year 2022. Cohort size is represented by circle size. N = number of patients.

**Figure 3 cancers-15-00174-f003:**
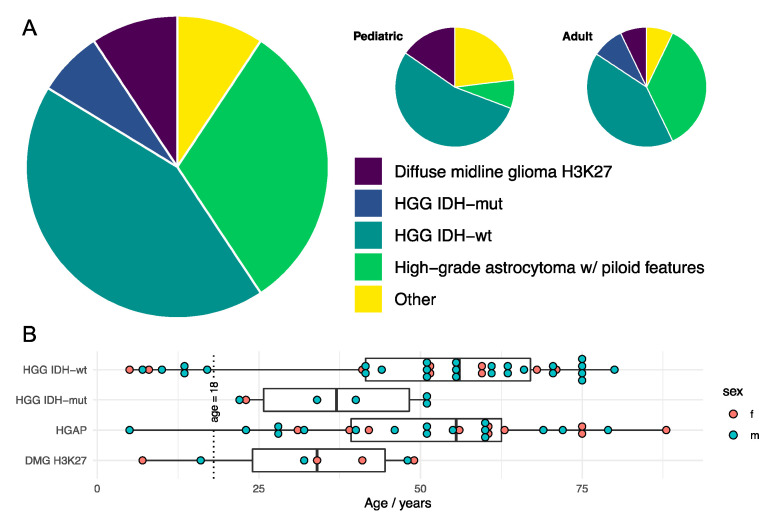
Molecular integrated diagnosis of histological cerebellar high-grade glioma. (**A**) Example proportions of subclass diagnosis of histological cerebellar high-grade glioma following integration of advanced molecular analyses. (**B**) Example age distribution of patients by integrated tumor diagnosis. Data from Reinhardt et al. [25].

**Table 1 cancers-15-00174-t001:** Comparative summary of cerebellar and supratentorial high grade glioma epidemiology.

Author	Study	Location	Male/%	Age/Years	MOS/Months
Babu et al. [9]	SEER database analysis	Supratentorial	59.5	μ = 61.8 *	7
		Cerebellar	58.1	μ = 56.6 *	8
Adams et al. [1]		Supratentorial	n.s. *p* = 0.87	μ (σ) = 61 (13) *	8 †
		Cerebellar		μ (σ) = 58 (16) *	9 †
Jeswani et al. [4]		Supratentorial	n.r.	7.4% < 40 *	8 †
		Cerebellar	62	23.5% < 40 *	7 †
Cho et al. [10]	Single cohort	Supratentorial	54.5	η = 55.3	16
		Cerebellar	64.1	η = 56.9	21
Picart et al. [5]	Single cohort	Supratentorial	60 *	μ (σ) = 63.2 (13.3) *	14 *
		Cerebellar	52.9 *	μ (σ) = 53.4 (15.7) *	6 *

* Significant difference between supratentorial and cerebellar. † Marginally significant difference calculated after adjusting for stratification variables. MOS = median overall survival, SEER = surveillance epidemiology and end results program, n.s. = not significant, n.r. = not reported, μ = mean, σ = standard deviation, η = median.

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
