# Peer review of "Cerebellar High-Grade Glioma: A Translationally Oriented Review of the Literature"

_cancers, 2022, doi:10.3390/cancers15010174_

Round 1

Reviewer 1 Report

In this manuscript, Raghu et al provide a review of literature on cerebellar high grade gliomas (cHGG), which focuses on the key topics of epidemiology, molecular characterization, and therapeutic approaches. The review is generally well organized, appropriately detailed, and clinically relevant. Some terminology needs to be updated and some additional explanations may be helpful. Overall, with minor revision, this review paper will be a strong candidate for publication.

Specific comments:

1) The primary/secondary GBM paradigm (lines 85-94) has fallen out of favor with the 2021 CNS WHO classification scheme, and the associated references (5, 6, 13) were all published prior to 2010. The authors describe the rarity of IDH mutation and EGFR amplification in cHGG as mixed primary and secondary features. It may be more accurate to describe them as features of IDH-wildtype GBMs in methylation classes “Midline” and “RTK-I” which make up a significant proportion of cHGG.

2) Based on current literature, are there any key genetic and/or epigenetic differences in H3K27M-altered midline gliomas occurring in the cerebellum versus the brainstem? This would be interesting to explore, if literature is available.

3) Please use Arabic numerals for CNS WHO grades rather than Roman numerals, as per 2021 CNS WHO guidelines (e.g., CNS WHO grade 4).

4) The correct CNS WHO terminology is “High grade astrocytoma with piloid features,” not pilocytic.

Author Response

Reviewer 1

In this manuscript, Raghu et al provide a review of literature on cerebellar high grade gliomas (cHGG), which focuses on the key topics of epidemiology, molecular characterization, and therapeutic approaches. The review is generally well organized, appropriately detailed, and clinically relevant. Some terminology needs to be updated and some additional explanations may be helpful. Overall, with minor revision, this review paper will be a strong candidate for publication.

Specific comments:

1) The primary/secondary GBM paradigm (lines 85-94) has fallen out of favor with the 2021 CNS WHO classification scheme, and the associated references (5, 6, 13) were all published prior to 2010. The authors describe the rarity of IDH mutation and EGFR amplification in cHGG as mixed primary and secondary features. It may be more accurate to describe them as features of IDH-wildtype GBMs in methylation classes “Midline” and “RTK-I” which make up a significant proportion of cHGG.

  • We thank the author for making this point. Although mentioning the primary/secondary paradigm remains relevant (at least for historical context), the reviewer is right we did not draw a clear enough line from these features to the ‘new’ IDH-wt subclasses. We have now revisited this paragraph to make this explicit and this has added more clarity to the narrative in this section.

2) Based on current literature, are there any key genetic and/or epigenetic differences in H3K27M-altered midline gliomas occurring in the cerebellum versus the brainstem? This would be interesting to explore, if literature is available.

  • This is indeed an interesting and relevant conjecture. Unfortunately, few groups who have studied significant series of H3K27M-altered midline gliomas report their epi/genetic alterations by tumour location. The San Francisco group and the Heidelberg groups have (see Solomon et al 2015 Brain Path, Schulte et al 2020 Neuro-onc Adv, and Ebrahimi 2019 J Can Res Clin Onc) but the cerebellar numbers are too small to point to any brainstem-cerebellar difference.
  • Looking at this question closer: comparing the K27-altered cases in cerebellum-specific series referenced in this manuscript with the non-cerebellar cases of the San Francisco or Heidelberg series does not reveal an obvious pattern of differences between cerebellar and other midline locations.

3) Please use Arabic numerals for CNS WHO grades rather than Roman numerals, as per 2021 CNS WHO guidelines (e.g., CNS WHO grade 4).

  • We thank the reviewer for pointing out this detail. We have now corrected this.

4) The correct CNS WHO terminology is “High grade astrocytoma with piloid features,” not pilocytic.

  • We appreciate the reviewer noticing this – we have corrected accordingly.

Reviewer 2 Report

It was not until recently that cHGG have started to be distinguished from sHGG in their epidemiological, mollecular and treatment option characteristics. The oriented review offered in this paper is clear, pertinent at this moment of cHGG ongoing research and treatment trials, and brings some understanding about pathogenesis, mollecular classification and future trends of treatment. The reference list shows a large and actual bibliographic revision. 

Author Response

Reviewer 2

It was not until recently that cHGG have started to be distinguished from sHGG in their epidemiological, molecular and treatment option characteristics. The oriented review offered in this paper is clear, pertinent at this moment of cHGG ongoing research and treatment trials, and brings some understanding about pathogenesis, molecular classification and future trends of treatment. The reference list shows a large and actual bibliographic revision. 

  • We thank the reviewer for their time and positive appraisal.

Reviewer 3 Report

In the manuscript titled "Cerebellar high-grade glioma: a translationally oriented review of the literature" prepared by Raghu et al., the author reviewed recent advances in cHGG classification, epidemiology, and therapeutic intervention. Overall I think the paper is well-organized and provide an interesting angle to this type of HGG. I have a couple of minor concerns as follow:

1. Could the author comment on the age-of-onset of cHGG? The author indicated that it is more frequent in younger patient. Are they still consider adult HGG, or pediatric disease?

2. The  cHGG may arise from inherited syndromes such as Li-Fraumeni. Did the author taken this into consideration?

3. The author claimed that IDH mutation is rare in cHGG. However, in Figure 3, the IDH-mutated case is only slightly less than H3K27M. Could the author provide the number of the percentage? Also combining with point 2, I have a feeling that the IDH-mutated cases are underestimated here. IDH mutation establishes very unique molecular and therapeutic profile (PMID: 32825279), which may indicate a different therapeutic strategy to this molecular subtype.  

Author Response

Reviewer 3

In the manuscript titled "Cerebellar high-grade glioma: a translationally oriented review of the literature" prepared by Raghu et al., the author reviewed recent advances in cHGG classification, epidemiology, and therapeutic intervention. Overall I think the paper is well-organized and provide an interesting angle to this type of HGG. I have a couple of minor concerns as follow:

  1. Could the author comment on the age-of-onset of cHGG? The author indicated that it is more frequent in younger patient. Are they still consider adult HGG, or pediatric disease?
  • The age-of-onset of cHGG is best appreciated through Table 1 and Figure 1A. Table 1 shows that the average age is younger (probably about 5 years) in cHGG vs. sHGG. It is also notable that the age-spread is slightly larger for cHGG, which points to a larger young-adult population (all these studies are adult only).
  • Fig 1A shows series that report a median age-of-onset for cHGG – some have excluded paediatric cases and some not – and seems to show that, as expected, paediatric cases are numerous enough to report a lower median age in the series that include them.
  • The paediatric cases should be considered paediatric disease. However, it remains very much unclear how paediatric and adult cHGG differ. For example, in the Reinhardt 2019 series where 13 of their 86 patients were paediatric (~15%), there was a higher proportion of integrated diagnosis of diffuse midline glioma K27, the only high-grade neuroepithelial tumor, and a lower proportion of high-grade astrocytoma with piloid features. Interestingly, the paediatric IDH-wt were mostly of the midline methylation subclass. However, while this may be a sentinel indicator of adult-paediatric differentiation, any firm conclusions are certainly premature due to the small case numbers. Nonetheless, to allude to a possible differentiation, we have now added sub-charts to figure 3 which give more detail on the Reinhardt 2019 data. We have also added a sentence to the manuscript summarising the above.
  1. Following point 1, the cHGG could arise from inherited syndromes such as Li-Fraumeni. Some of them also carry IDH mutation. Did the author taken this into consideration?

Yes, this is possible that cHGG could arise as part of an inherited syndrome, however this is not something that has been reported prominently in the series that we reviewed. Given the link between IDH1 mutations and Li Fraumeni astrocytomas this is certainly something worth exploring in cHGG but is not something the current literature can account for unfortunately.

  1. The author claimed that IDH mutation is rare in cHGG. However, in Figure 3, the IDH-mutated case is only slightly less than H3K27M. Could the author provide the number of the percentage? Also combining with point 2, I have a feeling that the IDH-mutated cases are underestimated here. IDH mutation establishes very unique molecular and therapeutic profile (PMID: 32825279), which may indicate a different therapeutic strategy to this molecular subtype.  

IDH mutation is uncommon with cHGG (see figure 2) occurring in about 8% of cases. Using the term ‘rare’ was an oversight and we thank the reviewer for pointing this out. We agree with the molecular-therapeutic perspective that the reviewer mentions with respect to IDH-mut tumours. Although, we think it is unlikely that IDH-mutated cases have been numerically greatly underestimated in our review of the literature. Figure 2 shows that 4 case series found no IDH mutated tumours, and the summary figure of ~8% is the best estimate attainable from the current published literature. However, to the reviewer’s point, it is true that most of these series did not test for rare (based on ‘supratentorial standards’) IDH mutations, so the prevalence of those in greater abundance cannot be excluded. Nonetheless, if such ‘rare’ mutations were common in cHGG the meaning would still be unclear - this could even point to molecular-therapeutic divergence from sHGG IDH-mut. We have now mentioned in the manuscript that rare IDH mutations were usually not tested for to add context. Ultimately, we have not focused on cHGG IDH-mut subtype in this review because at present there is not substantive evidence (yet) that it differs meaningfully from the supratentorial counterpart.